# Serpentoviruses Exhibit Diverse Organization and ORF Composition with Evidence of Recombination

**DOI:** 10.3390/v16020310

**Published:** 2024-02-18

**Authors:** Steven B. Tillis, Robert J. Ossiboff, James F. X. Wellehan

**Affiliations:** Department of Comparative, Diagnostic, and Population Medicine, College of Veterinary Medicine, University of Florida, Gainesville, FL 32608, USA; rossiboff@ufl.edu (R.J.O.); wellehanj@ufl.edu (J.F.X.W.J.)

**Keywords:** next-generation sequencing, Illumina MiSeq, nidovirus, recombination, reptile, RNA virus, snake

## Abstract

Serpentoviruses are a subfamily of positive sense RNA viruses in the order *Nidovirales*, family *Tobaniviridae,* associated with respiratory disease in multiple clades of reptiles. While the broadest viral diversity is reported from captive pythons, other reptiles, including colubrid snakes, turtles, and lizards of captive and free-ranging origin are also known hosts. To better define serpentoviral diversity, eleven novel serpentovirus genomes were sequenced with an Illumina MiSeq and, when necessary, completed with other Sanger sequencing methods. The novel serpentoviral genomes, along with 57 other previously published serpentovirus genomes, were analyzed alongside four outgroup genomes. Genomic analyses included identifying unique genome templates for each serpentovirus clade, as well as analysis of coded protein composition, potential protein function, protein glycosylation sites, differences in phylogenetic history between open-reading frames, and recombination. Serpentoviral genomes contained diverse protein compositions. In addition to the fundamental structural spike, matrix, and nucleoprotein proteins required for virion formation, serpentovirus genomes also included 20 previously uncharacterized proteins. The uncharacterized proteins were homologous to a number of previously characterized proteins, including enzymes, transcription factors, scaffolding, viral resistance, and apoptosis-related proteins. Evidence for recombination was detected in multiple instances in genomes from both captive and free-ranging snakes. These results show serpentovirus as a diverse clade of viruses with genomes that code for a wide diversity of proteins potentially enhanced by recombination events.

## 1. Introduction

Viruses of the subfamily *Serpentovirinae* are positive-sense RNA viruses documented in a variety of reptile species. In some instances, serpentoviruses can be primary pathogens associated with outbreaks of respiratory disease [1,2,3]. But serpentoviruses are also associated with chronic, subclinical infections that exhibit seasonal variations in prevalence [4,5,6]. While the greatest serpentoviral diversity is reported from numerous python species maintained in captivity [1,4,6,7,8,9], a wide variety of other reptile hosts have been documented, including invasive Florida pythons [5], free-ranging Chinese [10] and North American colubrid snakes [5], Chinese colubrid snake-associated nematodes [10], free-ranging Australian turtles [11], free-ranging Australian skinks [12], and captive chameleons [13]. The International Committee on Taxonomy of Viruses (ICTV) currently recognizes seven genera and twelve subgenera within the subfamily *Serpentovirinae*. The subfamily *Serpentovirinae* represents one of four subfamilies within the family *Tobaniviridae,* which is itself one of fourteen families within the order *Nidovirales.*

In addition to the aforementioned variability in clinical manifestations of serpentovirus infections, three isolated serpentoviruses showed strain-dependent 10^5^-fold differences in peak viral titer in in vitro infection assays on a python cell line [14]. Despite substantial sequence diversity across numerous open reading frames (ORFs) in serpentovirus genomes, thus far, genomic characterization has been primarily focused on phylogenetic relationships of the ORF1b gene [5,6], which encodes the evolutionarily conserved RNA-dependent-RNA-polymerase (RDRP), along with other replication-complex associated proteins [9].

Most RNA viruses have no replication proofreading, resulting in an error rate that limits their complexity and size. Due to a crude error checking mechanism exonuclease domain (ExoN) associated with their RDRP, viruses in the order *Nidovirales* can have some of the largest known RNA genomes [15]. Nidovirales genomes can be over 30,000 base pairs (bp) in length, as is seen in some serpentoviruses [13]. Genomes are primarily split into two main regions: ORF1ab coding for non-structural, replication associated, polyprotein 1ab (PP1ab) and a second region containing a variable number of nested, overlapping ORFs coding for structural or accessory proteins [15,16]. The structural proteins required for virion formation always include the spike protein (S), nucleoprotein (N), and matrix protein (M); however, additional structural proteins may also be required for the production of infectious virions [15,16]. Nonstructural accessory proteins may also play critical roles in viral replication [17,18].

The most well-characterized nidoviruses are those in the family *Coronaviridae.* For example, severe acute respiratory syndrome coronavirus 2 (SARS-CoV2) has four ORFs that code for four structural proteins and six ORFs encoding eleven accessory proteins, all in addition to ORF1ab [19]. The processes of infection and viral reproduction are complex, and the direct roles these proteins play in pathogenicity remains an area of research [17]. However, coronaviruses are better characterized and can be used as a model for serpentovirus viral reproduction. The spike protein permits virion binding to host cellular receptors, and, sometimes along with glycoproteins such as hemagglutinin-esterase (HE), initiates endocytosis into the cell [18]. Once inside the cell, the genome is released and translated by host cell free-ribosomes as cellular messenger RNA (mRNA) [19,20]. In addition to secondary structure originating from the RNA genome itself, 3′ and 5′ untranslated regions (UTRs) contain highly conserved stem and loop structures [21]. These secondary structures allow for circularization of the genome, enabling translation [21]. The first translated protein, PP1ab, forms the replication-transcription complex [19,20]. These replicative proteins begin transcribing smaller negative-sense (−) sub-genomic RNA covering only a subset of ORFs from the positive-sense (+) viral genome [19,20]. These nested sub-genomic RNAs are where the name *Nidovirales* (“nested viruses”) originates. The sub-genomic (−) RNA is then transcribed back into (+) mRNA which is used to produce viral proteins with host ribosomes [19,20]. Some viral proteins are translated in the cytoplasm, while membrane-bound proteins are translated in the endoplasmic reticulum [18,19,20]. Through poorly characterized mechanisms, structural proteins coalesce with nucleoprotein-bound full length viral genomes within Golgi vesicles, and release mature virions are released from the cell through exocytosis [19,20].

Coronaviruses also deploy other tools to enhance infection. Alternative splicing of coronaviral mRNA can aid in infection through a dysregulation of normal splicing of host genes [22,23,24,25]. Additionally, genome recombination is documented in coronaviruses at varying rates and is proposed as critical for maintaining viral diversity [26,27,28]. In vitro knock-out of the proofreading coronaviral ExoN resulted in fewer recombinants, indicating an additional function for this protein supporting viral recombination [26]. Coronaviral protein glycosylation varies between different organ systems as well as by host species, and represents another virulence factor [29]. Changes in glycosylation not only affect host range by changing receptor binding, but also aid in immune evasion [29,30,31].

The aims of this study are first, to characterize the genomes of eleven novel serpentoviruses, and second, to place the viruses in the wider context of the subfamily *Serpentovirinae* through a number of modeling techniques. Serpentovirus diversity was characterized by examining genome templates, protein composition, potential protein function, protein glycosylation sites, differences in phylogenetic history between ORFs, and recombination analysis between serpentovirus clades.

## 2. Materials and Methods

### 2.1. Sample Origin and Diagnostic Screening

Ten novel serpentovirus genomes generated for this study originated from diagnostic samples representing eight captive python species originating from multiple collections within the United States submitted to the Zoological Medicine Diagnostic (ZMDx) Laboratory at the University of Florida. Samples subjected to rtPCR included extracted RNA from both oral tissue and oral swabs. One additional novel genome was generated from cell culture lysate extraction of isolated green tree python serpentovirus (GTSV) as previously described [14]. Upon receipt, samples were extracted immediately using the Zymo Quick-RNA MiniPrep Kit (Zymo Research, Irvine, CA, USA), per manufacturer’s recommendation. Diagnostic samples were subjected to a modified rtPCR protocol as previously described using sense primer BarniPVTF and antisense primer BarniDYTR [6]. The rtPCR mix for each primer pair included 4 µL of 10 µM forward/sense primer, 4 µL of 10 µM reverse/antisense primer, 25 µL of 2× PCRBio rt-PCR mix, 11.5 µL of H_2_O, 2.5 µL of 20Xrtase Taq, and 3 µL RNA extract. Reactions were run in a MJ Research PTC-100 Thermal Cycler with conditions for each as follows: 50 °C for 10 min; 94 °C for 2 min; 94 °C for 30 s, 46 °C for 30 s, and 72 °C for 30 s for 40 cycles; and 72 °C for 7 min followed by holding at 4 °C. rtPCR products were visualized on a 1% agarose gel, and bands of approximately 150 base pairs (bp) in length were excised. Nucleic acids were extracted using Zymo Clean Gel DNA Recovery Kit (Zymo Research) per the manufacturer’s recommendation. Samples were submitted for bidirectional Sanger sequencing (Genewiz, South Plainfield, NJ, USA) and were edited and aligned using Geneious Prime (Version 11.0.18, Auckland, New Zealand); samples were considered positive if a serpentovirus sequence was returned as the closest match on NCBI BLASTX [32].

### 2.2. Illumina Next-Generation Sequencing

To capture serpentoviral genomic variation, samples were subjected to Illumina MiSeq next-generation sequencing. Previously extracted RNA was concentrated using a Zymo RNA Clean and Concentrator kit (Zymo Research). Ribosomal RNA was depleted using the NEBNext rRNA Depletion Kit (Human/Mouse/Rat) (NEBNext, Ipswich, MA, USA) according to manufacturer’s recommendations supplemented with AMPure XP beads (NEBNext). Libraries were generated using NEBNext Ultra II RNA Library Prep kit (NEBNext) using the manufacturer’s protocols. Sample libraries were pooled and loaded into an Illumina 600 cycle V3 MiSeq cartridge (Illumina Inc., San Diego, CA, USA) and run on an Illumina MiSeq system. Contigs of generated MiSeq reads were assembled De Novo in CLC Genomics Benchtop software (Version 20, CLC BIO). Sequences generated from the project were deposited in the Sequence Read Archive (SRA) and GenBank database. Assembled genomic sequence generated from Sanger and Illumina MiSeq sequencing can be found in Genbank accession numbers [OR131594-OR131648] and raw reads from Illumina MiSeq sequencing in Sequence Read Archive accession number [PRJNA982273]. Fragmented genomes were mapped to their closest existing published genomes determined by NCBI BLASTN and putative gaps were mapped in a composite alignment sequence using Geneious Prime. The GenBank accession numbers, host species, reference genome used to map genome gaps, number of sequenced base pairs, and the estimated portion represented by novel genomes are included in Appendix A. Assemblies of fragmented genomes can be found at [33]. 

### 2.3. Filling Genome Gaps with rtPCR

While the majority of extracted RNA from diagnostic samples was used in generating MiSeq libraries, extracted RNA remained for three samples; rtPCR was used to close gaps between generated genome fragments. Samples were subjected to an rtPCR process similar to the initial diagnostic PCR except for the annealing temperature and primer pair. The target virus, primer pair sequence, amplicon size, and annealing temperature used to close genome gaps are shown in Appendix A. Determined gap sequences were mapped onto genomic fragments using Geneious Prime.

### 2.4. 3′ and 5′ Rapid Amplification of cDNA Ends

Rapid amplification of cDNA ends (RACE) was performed for Bredl’s python serpentovirus [OR131621-622]. FirstChoice RLM-RACE Kit (Thermo Fisher, Waltham, MA, USA) was used with virus specific primers using the manufacture’s protocol. For 5′ RACE heminested primers were used to amplify unknown regions and included kit 5′ outer primer with primer BR5RACE-A (5′-AGG CTG GCT AAG TAG TTG TGC-3′) and kit 5′ inner primer and primer BR5RACE-B (5′-AGT AGT TGT GCT GGT ACA CTA AT-3′). For 3′ RACE heminested primers were used to amplify unknown regions and included kit 3′ outer primer with primer BR3RACE-C (5′-GAG ACC CTG AAC ACC ACT GG-3′) and kit 3′ inner primer with primer BR3RACE-D (5′-CCA CTG GGT CGT TTC CTC AT-3′). RACE amplicons were then Sanger sequenced to complete the terminal ends of the genome.

### 2.5. Genome Organization

In addition to the eleven novel genomes generated from this study, 57 previously published serpentoviruses and 4 related outgroup nidoviruses were used in all genomic and phylogenetic analyses where applicable genome portions were available. GenBank accession numbers for these viruses can be viewed in Figure 1 and Appendix A.

Using both published genome annotations and Geneious Prime open-reading-frame (ORF) finding tools, putative ORFs were annotated for each genome and translated sequence for each ORF was used for analysis. Easily recognized ORF motifs such as ORF1a, ORF1b, and ORF2 (Spike protein, S) were grouped together, and the remaining translated ORFs were aligned via fast Fourier transform (MAFFT) [34]. The next most common proteins, Nucleoprotein (N) and Matrix protein (M), were grouped together while the alignment process was repeated for remaining, ungrouped ORFs. The grouping process was repeated with the next most common protein while the alignment process continued for remaining ungrouped ORFs. These successive rounds of alignments and grouping continued until no more homologous ORFs between viruses could be identified. While some ORFs that shared homology between serpentovirus genera were given a common name and labeled with the ORF number as a subscript as needed to aid in clarity, other ORFs that were unique to clades were labeled as viral protein (VP) followed by the protein kilodalton weight and the ORF number as a subscript.

The next step of the analysis was to attempt to determine putative functions for each ORF via multiple methods. Amino acid sequences were first submitted to the Predict Protein server (https://PredictProtein.org, accessed on 3 April 2023) to examine membrane topology and binding domains [35]. Next, protein sequence similarity examination was performed on the HMMER web server (https://www.ebi.ac.uk/Tools/hmmer/, accessed on 3 April 2023) to compare against other characterized proteins [36]. E-scores less than 0.05 were considered statistically significant as a high-similarity protein match, while E-scores between 0.10 and 0.05 were considered a low-similarity match. E-scores greater than 0.10 were not considered as similar proteins. If no similar proteins or potential function could be determined, amino acid sequence was submitted to the ITASSER server (https://zhanggroup.org, accessed on 3 April 2023) for structural modeling [37,38,39]. Sequences were also screened for N-linked glycosylation sites using the NetNGlyc-1.0 server (https://services.healthtech.dtu.dk/services/NetNGlyc-1.0/, accessed on 3 April 2023) [40]. For proteins with multiple predicted N-linked glycosylation sites, only the site with the highest likelihood was reported in the analysis.

The 5′ and 3′ UTRs were also examined. Secondary stem and loop structure were modeled with alpha, beta, gamma, and delta coronavirus templates on the R2DT server (https://rnacentral.org/r2dt, accessed on 3 April 2023) [41].

### 2.6. Phylogenetic Analysis

Both novel and previously published serpentovirus genomes and outgroup genomes were subjected to phylogenetic analysis across common ORF1b, S, M and N ORFs. GenBank accession numbers for viruses used in phylogenetic analysis can be viewed in Figure 1 and Appendix A. All ORFs were translated, and the amino acid sequences were aligned using MAFFT [42]. Putative gaps in amino acid sequence for fragmented genomes were filled with an “X” as ambiguity. A Bayesian method of phylogenetic inference (Mr. Bayes 3.2.7a with gamma distributed rate variation, 4 chains of 2.5 × 10^6^ generations with 25% burn-in) for each ORF was performed on the CIPRES server [43,44,45]. Phylogenetic trees were visualized using FigTree software (http://tree.bio.ed.ac.uk/software/figtree/) (accessed on 8 January 2021). Viruses not yet recognized by the ICTV were given preliminary assessment of classification by pairwise uncorrected distances (PUD) analysis. Identity between amino acid sequences was examined from partial alignments of the ORF1b gene starting at the junction of pp1a/b to DEAD-like helicase C domain (1380 amino acids in length for BPNV1: KJ541759). Viruses were classified using ICTV previously proposed serpentovirus classification cutoffs of family (≥25.6–31.9% homology), subfamily (≥36.1–42.3% homology), genus (≥45.0–58.1% homology), subgenus (≥90.2–93.7% homology), and species (≥95.6–97.0% homology) [46].

### 2.7. Recombination Analysis

Viral recombination analysis was performed with the Recombination Detection Program (RDP4) for the following ORFs: ORF1a, ORF1b, S, M and N [47]. Nucleotide alignments for each ORF were aligned using MAFFT. After alignment putative nucleotide gaps were removed to allow the RDP4 software to run. Recombination analyses performed include the Recombination Detection Program (RDP) method (window size of 30) [48], a Bootscan analysis (window size of 200, step size of 20, replicates of 100, random number seed of 3, and cutoff percentage of 70) [49], and a MaxChi analysis (variable sites per window of 70) [50].

## 3. Results

### 3.1. Genome Organization

MiSeq NGS successfully generated large portions of genomic sequence (estimated 48–97% of genomes represented, Appendix A) for 11 novel viruses included in this study that fit within existing serpentovirus genera clades. The genome organization of these novel viruses was included along with 57 other serpentoviruses and four outgroup viruses to create a genome template for serpentoviral clades (Figure 2, Appendix A).

Serpentovirus genomes show diversity in both sequence and protein composition, with genome lengths varying from approximately 27,000 base pairs (bp) to 36,000 bp (Figure 2). Of the nine serpentovirus genera currently recognized by the ICTV, the shortest genomes belonged to the genus *Septovirus,* and the longest is Veiled Chameleon Serpentovirus B in the genus *Vebetovirus*. The genus *Sectovirus* is comprised of only two partially sequenced genomes, which lack the 5′ UTR sequence necessary to determine genus genome lengths.

While the length of serpentovirus genomes varied, most genera had a consistent genome structure organization (Figure 2, Appendix A). Untranslated regions are positioned at the 5′ and 3′ ends of each virus. The most 5′ portion of the genomic coding region consists of non-structural polyprotein 1ab (pp1ab). The second portion of the genome contains structural and accessory proteins. Although the composition of ORFs in this latter region is variable, at a minimum, all serpentoviruses contain S, M and N structural proteins (Figure 2, Appendix A).

Both the 5′ and 3′ UTR are predicted to utilize RNA secondary structure to circularize the genome, enabling ribosomal translation. Based on coronavirus templates, models of serpentovirus 5′ UTRs predicted five small stem and loop structures using *Betacoronavirus* 5′ UTR templates in R2DT (Appendix A). The length of the 5′ UTR varied from 289 bp up to 2271 bp for corn snake infratovirus [MZ971343], although complete terminal ends were not determined for all viruses. In contrast, models of serpentovirus 3′ UTRs predicted four small stem-loops and a fifth larger looped RNA using *Gammacoronavirus* template 3′ UTR in R2DT (Appendix A). A subset of genera also formed viable models using *Alphacoronavirus* template 3′ UTR (Appendix A), forming two small and one large stem-loop structures. Lengths for this region varied from 165 bp to 1653 bp in length for emerald tree boa sertovirus [MN161561].

Following the 5′ untranslated region, translation begins at ORF1ab coding for pp1ab. At 13530 to 19713 bp in length, ORF1a is the longest ORF. Apart from the genus *Septovirus*, ORF1a terminates with the slippery sequence 5′-AAAAAAC-3′ that enables a [−1] frameshift for the translation of ORF1b to begin (Figure 2). While the novel blood python septovirus [OR131603-605] still contains this slippery sequence, remaining septoviruses uniquely lack this sequence. Instead, the ORF1a coding region terminates, leaving an 8 bp gap before the start of ORF1b (Figure 2). ORF1b is the second largest ORF at 6516 to 7086 bp in length.

The predicted S_(2)_ ORF follows ORF1b, and demarcates the start of structural and accessory protein ORFs in the genome. For most serpentoviruses, the start of the S_(2)_ ORF (2454 to 3618 bp, ≥17% amino acid [aa] homology) is nested within the end of ORF1b, although this is not the case for *Vebetovirus*, or some members of *Lyctovirus* and *Pregotovirus* (Figure 2).

In seven genera, the ORF that follows S_(2)_ is a predicted transmembrane/minor matrix protein 1 (TM1_(3)_), with the exception of both veiled chameleon serpentoviruses (in *Vebetovirus* and *Lyctovirus*; Figure 2, Appendix A). For most genera the TM1_(3)_ ORF (546 to 1440 bp, ≥5% aa homology) is nested at the end of the S ORF. Predicted membrane topology for this protein includes an extracellular portion sandwiched between two transmembrane helixes with predicted N-linked glycosylation sites (73%). Although this region is variable, no matches were found to existing characterized proteins (Table 1).

After S_(2)_, TM1_(3)_ (if present), and some unique proteins in some genera, a predicted M_(3/4/5)_ ORF (567 to 846 bp, ≥11% aa homology) is followed immediately by the predicted N_(4/5/6)_ ORF (387 to 741 bp, ≥10% aa homology), except for the bovine torovirus outgroup [LC088094], which contains a predicted glycoprotein between M_(3)_ and N_(5)_ (Figure 2, Appendix A). Although for most serpentoviruses the 5′ end of the N ORF is nested within the M ORF, this feature was absent in *Infratovirus* and some members of *Pregotovirus* (Figure 2). For the simplest serpentovirus genomes, belonging to the genus *Sertovirus*, the N_(5)_ ORF is the fifth and final ORF before the 3′ UTR (Figure 2, Appendix A).

After *Sertovirus*, the next simplest serpentovirus genomes belong to the genus *Infratovirus*. This genus had an additional ORF4 between TM1_(3)_ and M_(5)_ ORFs. Predicted *Infratovirus* ORF 4 protein variation includes VP7b_4_, VP10_4_, and VP13_4_ (180–387 bp, ≥17% aa homology). Predicted topology for VP7b_4_ and VP13_4_ includes a single predicted transmembrane helix with an extracellular portion ending in a signal peptide; protein VP10_4_ lacked predicted membrane topology (Table 1). Protein VP7b_4_ had a significant high-similarity match (E-value 0.0002, 60.6% similarity) to proteins associated with a globin gene transcription factor while VP13_4_, sequenced from a snake nematode, had a significant high-similarity match (E-Value 1.7 × 10^−5^, 52.2% similarity) to an uncharacterized nematode protein from the hookworm *Ancylostoma ceylanicum* (Table 1). None of the *Infratovirus* ORF4s had predicted N-linked glycosylation sites.

*Septovirus* genomes are also relatively simple. Septoviruses have a similar genome layout to five-ORF *Sertovirus*, but contain an additional ORF6 (255–330 bp, ≥20% aa homology). If the novel blood python *Sertovirus* [OR131603-605] VP8_6_ is excluded, 50% amino acid homology is shared for ORF6 in remaining members of this genus. Neither VP8_6_ nor VP13_6_ had any close protein matches, predicted membrane topology, or predicted N-linked glycosylation sites, although they were predicted to contain protein binding domain regions (Table 1). 

The genus *Pregotovirus* has the largest number of sequenced genomes, all containing seven ORFs. In addition to S_(2)_, TM1_(3)_, M_(4),_ and N_(5)_, pregotoviruses have a predicted transmembrane/minor matrix protein 2 (TM2_(6)_) (767 to 1434 bp, ≥8% aa homology) as ORF6, and glycoprotein GP1_(7)_ (1440 to 1533 bp, ≥13% aa homology) as ORF7 (Figure 2, Appendix A). Both TM2_(6)_ and GP1_(7)_ have predicted membrane topology that includes a transmembrane helix with extracellular portions terminating in a signal peptide. TM2_(6)_ has predicted N-linked glycosylation sites (77%) and a low-similarity match (E-Value 0.051, 56.9% similarity) to a protein involved in cell surface chemotaxis and apoptosis regulation (Table 1). GP1_(7)_ has predicted N-linked glycosylation sites (70%) but no close protein matches (Table 1).

The genus *Sectovirus* has a genome organization that includes eight ORFs. In addition to S_(2)_, TM1_(3)_, M_(4),_ N_(5)_, and GP1_(7)_, sectoviruses have VP38_4_ or VP43_4_ as ORF4 and VP37_8_ as ORF8 (Figure 2, Appendix A). VP38_4_ (1017 bp) and VP43_4_ (1146 bp) share 43% amino acid homology and predicted N-linked glycosylation (75% and 77%, respectively). Predicted topologies for VP38_4_, VP43_4_, and VP37_8_ all include a single transmembrane helix with extracellular portions terminating in a signal peptide. VP38_4_ has a significant match (E-Value 5.5 × 10^−6^, 62.5% similarity) with an uncharacterized *Anolis* lizard protein, while VP43_4_ had a significant match (E-Value 0.0051, 47.7% similarity) with a peroxidase enzyme from a non-domesticated rice (*Oryza barthii*) (Table 1). At the end of the genome, VP37_8_ is (987 bp, ≥54% aa homology) has predicted N-linked glycosylation (70%) but no similar protein matches (Table 1). GP1_(7)_ from the genus *Sectovirus* has between 15 to 18% aa homology to the GP1_(7)_ from *Pregotovirus*, although they still clustered together in protein alignments. Within the genus *Pregotovirus,* homology as low as 36% aa was observed within GP1_(7)_.

The genus *Lyctovirus* includes two North American watersnake viruses with only six ORFs (Figure 2). Related Chinese colubrid viruses also in the genus *Lyctovirus* have a similar genome organization, except for the insertion of an additional 201–354 bp ORF4 between the TM1_(3)_ and M_(5)_ ORFs. The two proteins encoded by ORF4, VP7_4_ and VP13_4_, (34% aa homology) are predicted to contain one or two predicted transmembrane helixes, respectively. VP7_4_ has a significant match (E-Value 1.1 × 10^−5^, 51.1% similarity) with a protein from the Japanese scallop (*Mizuhopecten yessoensis*) involved in cytoskeleton reorganization and transportation, with no predicted N-linked glycosylation sites (Table 1). VP13_4_ also has a significant match (E-Value 2.8 × 10^−17^, 75% similarity) with a fungal (*Thanatephorus cucumeris*) tetratricopeptide repeat domain involved with multi-protein scaffolding and predicted N-linked glycosylation sites (71%) (Table 1). Regardless of the host continent of origin, snake lyctoviruses all terminated with glycoprotein GP2_(6/7)_ (2028–2202 bp, ≥31% aa homology), which was unique in being separated by 85 to 130 bp from the previous ORFs. This is the largest gap observed between any serpentovirus ORFs. GP2_(6/7)_ has predicted N-linked glycosylation sites (77%) and a significant match (E-Value 0.0094, 47.5% similarity) with a putative immunoglobulin domain containing protein from the bacteria *Leucobacter komagatae* (Table 1). GP2_(6/7)_ did not cluster in organizational alignments with GP1_(7)_ seen in other serpentovirus genera, sharing only 7 to 11% aa homology with GP1_(7)_.

The most atypical serpentovirus genome organization was observed in veiled chameleon serpentoviruses. Veiled Chameleon Serpentovirus A [MT997160], currently in the genus *Lyctovirus*, subgenus *Chalatovirus*, has 8 ORFS including S_(2)_, VP54_3_, M_(4)_, N_(5)_, VP33_6_, VP12_7_, and VP7_8_ (Figure 2, Appendix A). VP54_3_ (1434 bp) has a single transmembrane helix with extracellular components and predicted N-linked glycosylation (78%). VP54_3_ has a low-similarity match (E-Value 0.053, 48.3% similarity) to an acyltransferase enzyme (Table 1). VP33_6_ (882 bp) has a predicted extracellular component sandwiched between two transmembrane helixes and predicted N-linked glycosylation (61%). It has a significant match (E-Value 0.03, 58.6% similarity) to a zinc finger domain containing protein from a filarid worm (*Loa loa*; Table 1). VP12_7_ (309 bp) has no predicted topology nor N-linked glycosylation sites and no close protein matches (Table 1). Lastly, VP7_8_ (180 bp) has no predicted topology but a predicted N-linked glycosylation site (70%). It has a significant match (E-Value 0.0014, 60.0% similarity) to Tobacco-Mosaic Virus resistance protein (Table 1).

The remaining serpentovirus genus, *Vebetovirus*, is currently a monotypic genus comprised of Veiled Chameleon Serpentovirus B [MT997159]. This virus has 6 ORFs including S_(2)_, VP12_3_, VP34_(4)_, M_(5)_, and N_(6)_ (Figure 2, Appendix A). VP12_3_ (327 bp) has an extracellular component sandwiched between two transmembrane helixes and no predicted N-linked glycosylation sites. It has a significant match (E-Value 0.00071, 58.1% similarity) to a bacterial (*Shewanella* sp.) protein involved with calcium binding and expulsion (Table 1). Lastly, VP34_4_ (939 bp) has no predicted topology or similar protein matches, but did contain predicted N-linked glycosylation sites (75%) (Table 1).

### 3.2. Phylogeny of Novel Viruses

The phylogenetic analysis of the ORF1b gene for the eleven novel python serpentovirus genomes generated in this study compared to previously published serpentoviruses is shown in Figure 1. Nine of the novel genomes fit into the existing *Pregotovirus* clade, although two of the genomes represent more divergent taxa within the genus. The PUD analysis supports the designation of novel subgenus and species for reticulated python serpentovirus 2 (Retic-V2) (Host: *Malayopython reticulatus*, ≤89.9% similar) [OR131618-620] and Papuan python serpentovirus (Host: *Apodora papuana*, ≤88.2% similar) [OR131606-607]. There was support for designation of species for white lip python serpentovirus (Host: *Bothrochilus albertisii*, ≤94.1% similar) [OR131594-600]. The PUD analysis did not support the designation of novel species for diamond python serpentovirus (Host: *Morelia spilota spilota*, ≤98.2% similar) [OR131642-648], Bredl’s python serpentovirus (Host: *Morelia bredli*, ≤96.2% similar) [OR131621-622], green tree python serpentovirus (Host: *Morelia viridis*, 95.7% similar) [OR131641], black headed serpentoviruses (Host: *Aspidites melanocephalus*), ≤97.8% [OR131631-640] and ≤97.1% [OR131623-630] similar), or green tree python serpentovirus isolate (GTSV) (Host: *Morelia viridis*, ≤100% similar).

An oral swab sample from a captive reticulated python suffering from respiratory disease resulted in the amplification of two coinfected serpentoviruses. While one virus (Retic-V2) has support by PUD analysis as a novel species and subgenus within *Pregotovirus*, the other virus (Retic-V1) [OR131608-617] has support as the same viral species (97% similar) as the existing reticulated python septovirus [MN161566]. There was inequivalent NGS RNA sequence coverage for the two reticulated python viruses in the sample. For Retic-V1 13,184 bases of the genome were amplified at an average coverage of 6.68. Retic-V2 was sequenced at almost six times the coverage, amplifying 31,475 bases of the genome at an average coverage of 38.73.

Another divergent novel serpentovirus was found in a sample from a blood python (*Python brongersmai*) [OR131603-605]. The sequence of this virus most closely matches serpentoviruses in the genus *Septovirus* (Figure 1). The classification as a novel species and subgenus has support by PUD analysis (70.3% similar to Burmese python clade 2 septovirus [MZ971286]).

### 3.3. Phylogenetic Incongruence

Phylogenetic trees of S, M, and N ORFs show phylogenetic incongruence (Figure 3), suggesting different evolutionary histories between ORFs. The phylogenetic tree for the spike protein has discrepancies from the ORF1b tree (Figure 3a). The genus *Lyctovirus* is split into two clades: one clade of Asian colubrid viruses which cluster with other Asian colubrid serpentoviruses in the genus *Sectovirus* with 100% posterior probability, and a second clade of remaining lyctoviruses which remains weakly clustered (58% posterior probability) with the genus *Vebetovirus* as seen in the ORF1b tree. Another discrepancy is the clustering of Bovine Nidovirus [NC027199] S protein with *Septovirus* genus serpentoviruses (91% posterior probability) instead of *Sertovirus* genus serpentoviruses as seen in the ORF1b tree and M trees.

The phylogenetic tree of the matrix protein also has incongruence with the ORF1b tree (Figure 3b). Veiled Chameleon Serpentovirus B in the genus *Vebetovirus* weakly clusters with the genus *Sectovirus* (serpentoviruses of Asian origin) (74% posterior probability) and the genus *Pregotovirus* (77% posterior probability), instead of clustering with the genus *Lyctovirus* as seen in other trees.

The phylogenetic tree of the N protein also has aberrations in the clustering of the genus *Pregotovirus* (Figure 3c). The majority of pregotoviruses weakly cluster with fish bafiniviruses and oncotshaviruses (71% posterior probability), which is not found in other trees. A lone, partially sequenced, Shingleback skink pregotovirus [KC184715] N protein is placed far from other pregotoviruses in a weakly supported clade (74% posterior probability) with the genus *Vebetovirus*. Finally, Bovine Nidovirus [NC027199] is placed in a weakly supported clade (53% posterior probability) with the genus *Lyctovirus*, which was not modeled in other trees.

For all phylogenetic trees reported in our study, bovine nidovirus [NC027199] clustered within the subfamily *Serpentovirinae* (Figure 2 and Figure 3), although this virus is not currently classified by the ICTV within this subfamily. PUD values determined for comparison of bovine nidovirus to the *Serpentovirinae* show homologies as low as 36% to 41% against the most closely related serpentoviruses of the genus *Sertovirus*. This PUD range supports that bovine nidovirus should be excluded from the *Serpentovirinae* (subfamily threshold: 36.1–42.3% homology). PUD values for the genus *Sertovirus* support its current inclusion in *Serpentovirinae* (39.1% to 42.5% similar).

### 3.4. Recombination Analysis

No evidence of recombination is detected within the ORFs M, N or S, but recombination events were supported in both ORF1a and ORF1b.

A green tree python pregotovirus [MK182566] is predicted as a recombinant of major-parent, closely clustered, green tree python pregotovirus [MN161559] and a more distant blood python pregotovirus [MN161564] for portions of ORF1a (Figure 4a). Recombination in this region are supported by RDP (*p*-value: 1.3 × 10^−160^), Bootscan (*p*-value: 4.7 × 10^−29^), and MaxChi models (*p*-value: 1.0 × 10^−47^).

In another green tree python pregotovirus [MK722373], a recombination event in ORF1a was supported between major-parent green tree pregotovirus [MN161559] and a GTSV isolate [OR131601-602] (Figure 4b). Recombination in this region was supported by RDP (*p*-value: 6.0 × 10^−85^), Bootscan (*p*-value: 4.6 × 10^−68^), and MaxChi models (*p*-value: 2.8 × 10^−46^).

A final incidence of recombination was detected in portions of the ORF1b of Burmese python septoviruses. Florida Burmese python virus clade 4 [MZ971330-MZ971340] is predicted to be a recombinant of closely clustered Burmese python virus clade 2 [MZ971286] with two regions from Burmese python virus clade 3 [MZ971304-MZ971305] (Figure 4c). Recombination in these regions was supported by RDP (2 regions, *p*-value: 3.6 × 10^−2^, 2.7 × 10^−13^), Bootscan (2 regions, *p*-value: 3.6 × 10^−4^, 2.8 × 10^−10^), and MaxChi models (1 region, *p*-value: 1.3 × 10^−11^).

## 4. Discussion

In this study, the genomes of eleven novel serpentoviruses are reported. By way of a number of modeling and phylogenetic techniques, the genomes of these novel viruses were compared with all other published serpentovirus genomes to characterize this rapidly growing family of predominantly reptile viruses.

Some genome attributes are consistent across serpentovirus genera. Where available, most serpentovirus 5′ UTRs could be modeled after Betacoronavirus 5′ UTRs, while 3′ UTRs could be modeled after Alphacoronavirus or Gammacoronavirus 3′ UTRs. While the specific functions of 3′ and 5′ secondary RNA structure in coronaviruses remain poorly characterized, high levels of conservation are likely necessary as secondary structure alterations can inhibit viral reproduction [21,51].

Another common attribute to serpentoviruses is that ORF1ab is always followed by the predicted spike protein, which for most serpentoviruses is followed by a predicted minor transmembrane protein (TM1_(3)_). A predicted matrix protein also immediately followed by the predicted nucleoprotein is also a feature common to all serpentovirus genomes. However, aside from these commonalities, serpentoviruses show wide genome variability in both ORF composition and amino acid homology.

Outside of the ubiquitous structural proteins S, M, and N, it is unclear without further in vitro experimentation if other serpentovirus ORFs serve as structural or accessory proteins. Many of the uncharacterized serpentovirus ORFs encode proteins having both membrane-associated topology and sites with a high likelihood of N-linked glycosylation. Proteins GP2_(6/7)_ and VP33_6_ have varying degrees of similarity to previously characterized membrane-associated glycoproteins, suggesting potential roles as structural proteins influencing membrane binding or immune recognition. The GP2_(6/7)_ protein has a high-similarity (E-Value 0.0094, 47.5% similarity) to a bacterial immunoglobulin domain-containing protein. A broad assemblage of viruses utilize similar immunoglobulin domains, including SARS-CoV-2 and related viruses [52], to fulfill a number of roles relating to pathogenicity and immune evasion [53].

A number of predicted serpentovirus proteins have high homology to proteins implicated in cellular apoptosis. Viruses utilize multiple strategies to inhibit apoptosis both to evade the immune system [54,55] as well as to paradoxically induce apoptosis to enable virion release [56,57]. Viruses in the closely related subfamily *Torovirinae* can induce apoptosis in epithelial cells [58,59,60]. As epithelial hyperplasia, occasionally with pyknotic cellular debris, is a common finding in numerous snakes infected with *Pregotoviruses*, including ball pythons and green tree pythons, apoptosis of epithelial cells may play an important role in viral pathogenesis [1,2,9,61]. *Pregotovirus* protein TM2_(6)_ has low-similarity (E-Value 0.051, 56.9% similarity) to a protein regulating apoptosis, while *Sectovirus* protein VP43_4_ has high-similarity (E-Value 0.0051, 47.7% similarity) to a peroxidase enzyme. The regulation of cellular hydrogen peroxide plays a critical role in regulating apoptosis [62,63,64]. Some viruses, such as *Human Betaherpesvirus 5* (HBHV5), subvert this mechanism to inhibit cellular apoptosis [65]. Similarly, *Lyctovirus* protein VP7_4_ shows significant similarity (E-Value 1.1e−05, 51.1% similarity) to MICAL-like protein 2 (MICAL-2), a host protein that plays a number of roles in diverse cellular processes, including regulating cellular cytoskeletal dynamics, exocytosis, and cell death [66]. MICAL-2 has been implicated in both intracellular protein trafficking and the prevention of cell death by inhibiting Ca2+ ion signals within mitochondria [67].

Other serpentovirus ORFs have high-similarity matches to proteins involved in multiprotein scaffolding (VP13_4_), transcription factors (VP7b_4_, VP38_4_), and enzymes (VP12_3_) with varying potential roles in viral replication. A final protein of note, VP7_8_ in Veiled Chameleon Serpentovirus A (*Lyctovirus*), exhibits significant similarity (E-value 0.0014, 60.0% similarity) to tobacco mosaic virus resistance protein (TMVRP). In infected tobacco plants, the binding of TMVRP to TMV helicase proteins causes conformational changes in TMVRP [68]. After said confirmational change, TMVRP interacts with down-stream signaling factors that trigger immune inhibition of viral replication [68]. VP7_8_ may act as a non-functional analog competing for binding sites on signaling factors triggered by a TMVRP-like molecule in reptiles. Such interactions could prevent signals to trigger the host immune response. HBHV5 also produces proteins that outcompete host proteins, subverting host immune responses in a similar manner [69].

Multiple discrepancies exist between the resultant trees of phylogenetic analysis of ORF1b, S, M and N. While it is possible that the analysis of shorter ORFs could limit the reliability of the results, some discrepancies may have plausible explanations supporting ORF-dependent evolutionary histories. The ORF1b tree supports the genus *Lyctovirus* as a monophyletic clade, but the S tree of the same genus splits *Lyctovirus* into two lineages of spike proteins. Instead, Asian colubrid lyctoviruses are grouped with other Asian colubrid-related viruses in the genus *Sectovirus*. This may indicate that the spike protein for these viruses may share a common origin for Asian colubrid species that is not shared with other lyctovirus S ORF or ORF1b portions of the same genomes, presumably due to recombination.

Phylogenetic incongruence was also noted in the phylogenetic analysis for ORF N. In this analysis the only documented lizard *Pregotovirus*, originally identified in free-ranging Australian shingleback skinks, loosely clustered with Veiled Chameleon Serpentovirus B, genus *Vebetovirus*. Unfortunately, further analysis of this phylogenetic relationship is limited to partial sequence of shingleback skink N ORF (107 aa) compared to Veiled Chameleon Serpentovirus B (170 aa). Similar to this example in serpentoviruses, papillomaviruses also have variable genomes that contain ORFs which may not share a common evolutionary origins [70]. All papillomavirus genomes contain five common ORFs but may also contain differing ‘adaptive protein’ ORFs with unique evolutionary origins [70]. Many papillomaviruses are observed to have narrow host ranges and often co-speciate with host species taxa [70]. It is hypothesized that the addition or loss of ‘adaptive protein’ ORFs are a factor that limits papillomavirus host range [70]. Likewise, in serpentoviruses the combination of specific ORFs with differing evolutionary origins could drive viral speciation and define the host range of serpentovirus lineages.

For the eleven novel serpentovirus genomes generated in this study, PUD analysis supports the establishment of four novel viral species as well as three novel subgenera existing within two current serpentovirus genera. However, this analysis alone does not fully represent the multi-faceted evolutionary relationships between serpentoviruses. For example, all serpentovirus genera followed a single genus-wide genome organization template except for *Lyctovirus*. Three lineages of *Lyctovirus* can be split into three unique genome organization templates. PUD analysis supports the designations of subgenera for these three lineages all under the genus *Lyctovirus*. Although ORF1b PUD analysis supports existing classification as three subgenera, differences in genome organization may warrant consideration to elevate current *Lyctovirus* clades as unique genera.

A final point of discussion regarding our phylogenetic analyses is the consistent placement of bovine nidovirus, genus *Bostovirus,* within the *Serpentovirinae* subfamily. The genus *Bostovirus* is currently classified by the ICTV in the subfamily *Remotovirinae,* family *Tobaniviridae*, along with three other subfamilies, including *Serpentovirinae*. The results of our phylogenetic tree analysis suggest that the genus *Bostovirus* could represent another genus in *Serpentovirinae.* Alternatively, our phylogenetic analysis could support the genus *Sertovirus* as the first representative of a new subfamily. However, genome template analysis may support current taxonomy, with *Sertovirus*-serpentoviruses sharing a common ORF (TM1_(3)_) with other serpentoviruses. Additional distinctions can also be made in both ORF size and genome organization between the genus *Bostovirus* and current serpentoviruses. Moreover, in additional support of the currently established ICTV taxonomy, PUD analysis was consistent with currently established genus and subfamily taxonomic designations. Regardless of evolutionary relationships, the conflicting taxonomic analyses of *Bostovirus* and current serpentoviruses serves as an example that defining viral evolution can be complicated, especially if different regions of viral genome do not share common origins.

One explanation for how different serpentovirus genome segments from distinct evolutionary origins are found within a single viral genome is recombination [71]. In the coronavirus murine hepatitis virus, up to 25% of viral progeny from samples co-infected with two strains showed evidence of recombination [27]. We detected multiple instances of recombination in this study. Serpentoviruses found in invasive, free-ranging Burmese pythons from Florida formed five clades within the genus *Septovirus*: Clade 1A, 1B, 2, 3, and 4 [5]. The discovered viral clades were locality dependent, however, and in one locality, all snakes were positive for only viruses in Clade 3 on initial sampling [5]. Unexpectedly, a single PCR negative snake from the Clade 3 locality converted to PCR positive a few weeks after capture. The virus detected in that python was divergent, did not closely match any other viruses detected in the study, and represented the sole Clade 4 virus detected [5]. The recombination analysis in this study suggests that the divergent Burmese python Clade 4 virus represents a recombinant primarily comprised of a Clade 2 virus with two regions of ORF1b replaced by Clade 3 viral sequences.

The analysis for this study was not comprehensive and did not include close examination of subdomains within ORF1ab. Two reptile nidoviruses were not included in this study including a pond slider nidovirus (Host: *Trachemys scripta*) which did not share recognizable serpentovirus genome organization, and a chameleon serpentovirus (Host: *Kinyongia boehmei*) with no publicly available sequence that clustered the closest to Veiled Chameleon Serpentovirus B in the genus *Vebetovirus* [72]. Some genomes contained large gaps limiting analysis for those genome regions, and the inclusion of multiple outgroups with the genera of *Bostovirus*, *Bafinivirus*, *Oncotshavirus*, and *Torovirus* may introduce error to phylogenetic tree models [73]. Although many proteins contained multiple predicted N-linked glycosylation sites, only the site with the highest likelihood was reported in the analysis. Finally, further in vitro and reverse genetics work would be needed to make a real determination of protein function.

Through characterization of eleven novel serpentoviral genomes followed by pan-serpentoviral phylogenetic analysis, this study explores the viral diversity of the rapidly expanding *Serpentovirinae* subfamily. Study findings suggest that serpentoviruses utilize a number of strategies to infect multiple reptile taxa. Serpentoviruses clades contain variation in ORF characteristics including composition, amino acid homology, potential evolutionary origin, potential translated protein function, and potential glycosylation sites. Support for recombination in serpentovirus lineages provides a mechanism to partially explain observed ORF diversity. While many specifics about serpentoviruses genomes remain a mystery, this study organizes and characterizes novel and published serpentovirus variation.

## Figures and Tables

**Figure 1 viruses-16-00310-f001:**
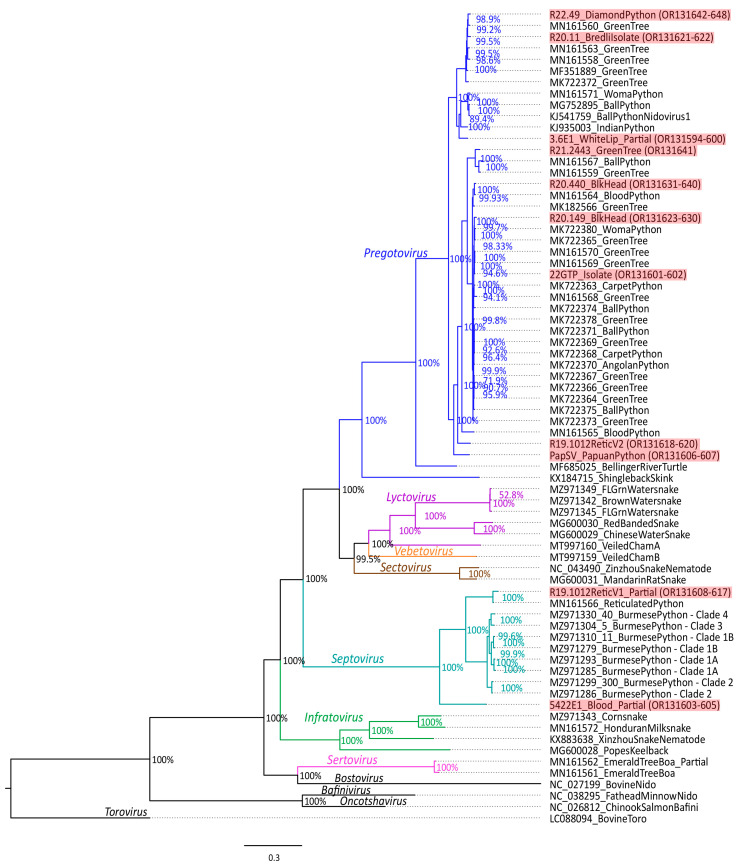
Phylogenetic tree of serpentovirus open reading frame 1b (ORF1b) amino acid sequence. Bayesian posterior probabilities are shown at branch points. Novel genomes reported in this study are highlighted in red. The seven currently recognized serpentovirus genera are label and color coordinated to corresponding branches (*Pregotovirus*: blue, *Lyctovirus*: purple, *Vebetovirus*: orange, *Sectovirus*: brown, *Septovirus*: teal, *Infratovirus*: green, *Sertovirus*: pink, outgroup taxa: black).

**Figure 2 viruses-16-00310-f002:**
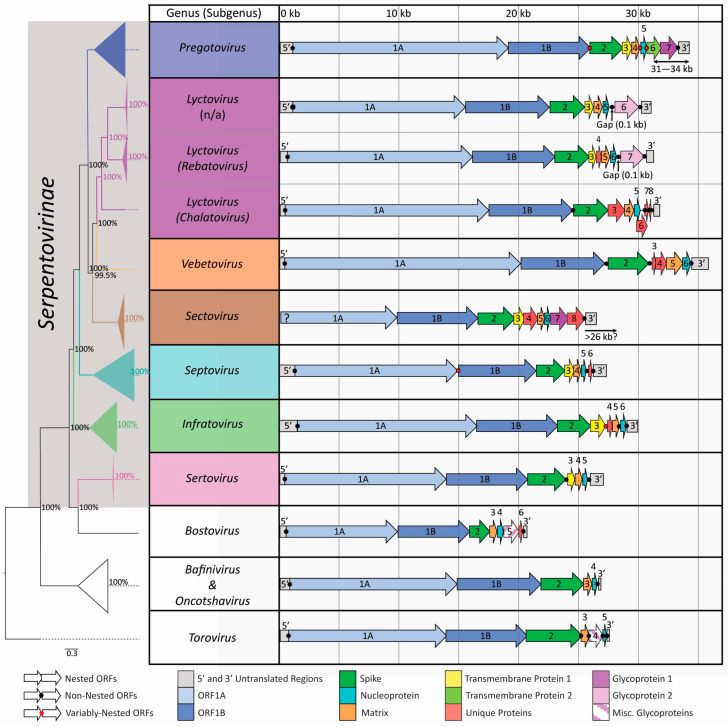
Genome organization and relatedness of the seven currently recognized serpentovirus genera: *Pregotovirus*, *Lyctovirus*, *Vebetovirus*, *Sectovirus*, *Septovirus*, *Infratovirus*, and *Sertovirus*. Phylogeny between clades is shown with a collapsed open reading frame b (ORF1b) amino acid tree. The viral subfamily *Serpentovirinae* is highlighted in gray. Genome templates were created by identifying potential Open Reading Frames (ORFs) and grouping common ORFs via Multiple Alignment using Fast Fourier Transform (MAFFT) alignments. Genome sections consist of colored arrows (ORFs) and gray boxes (untranslated regions) that represent the length of each genome section. Genome sections are separated by either a black dot (discrete ORF separation), red dot (variably nested ORF separation), or no dot (overlapping, nested ORFs).

**Figure 3 viruses-16-00310-f003:**
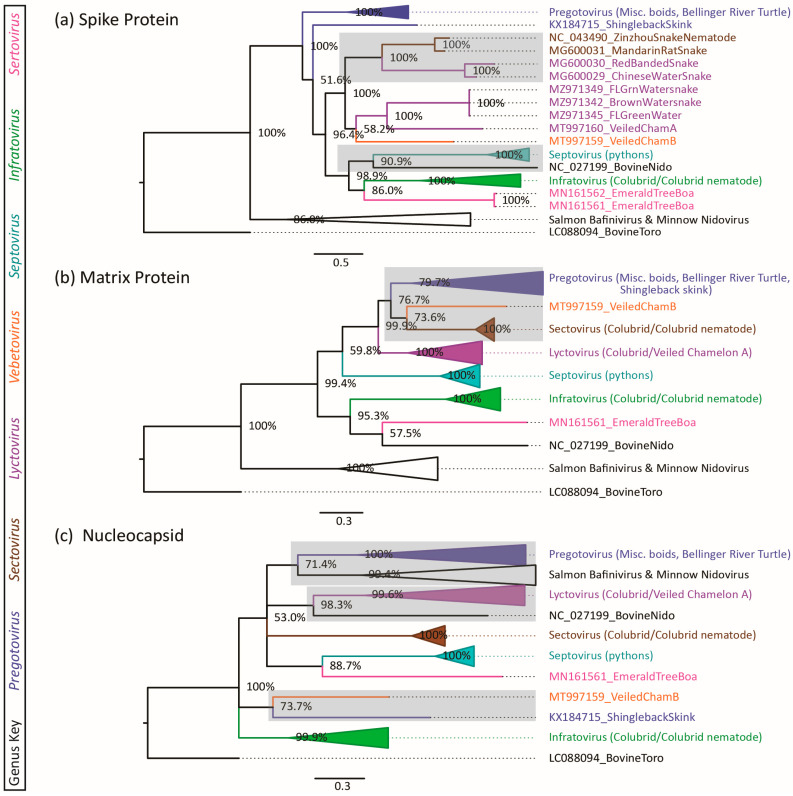
Phylogenetic trees of serpentovirus (**a**) spike, (**b**) matrix, and (**c**) nucleoprotein amino acid sequences. Bayesian posterior probabilities are shown at branch points. Viral clades are collapsed where appropriate. Areas of aberrant clustering different from ORF1b phylogeny are highlighted in grey. Serpentovirus genera are color-coordinated between trees and labeled (*Pregotovirus*: blue, *Lyctovirus*: purple, *Vebetovirus*: orange, *Sectovirus*: brown, *Septovirus*: teal, *Infratovirus*: green, *Sertovirus*: pink, outgroup taxa: black).

**Figure 4 viruses-16-00310-f004:**
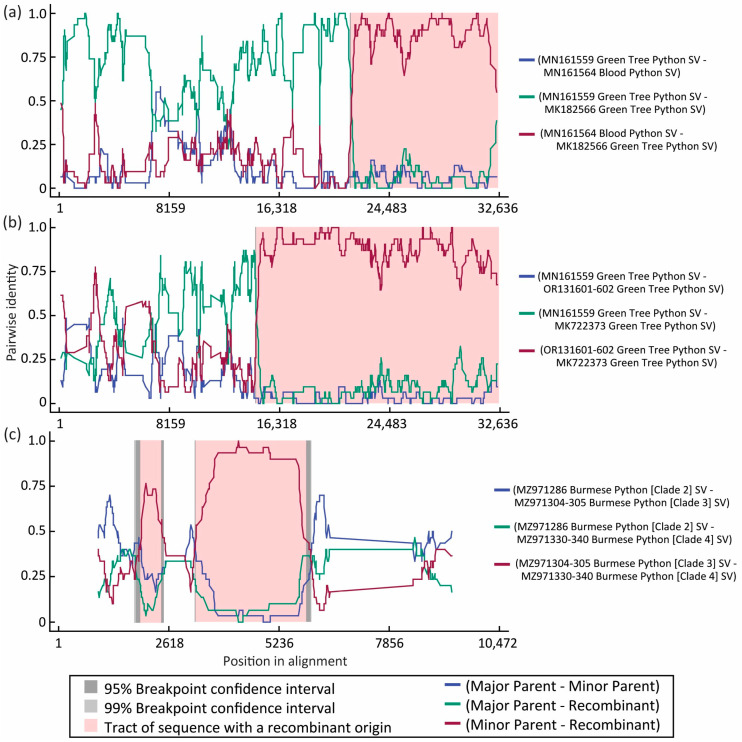
Graphs of genomes showing high support for recombination through Recombination Detection Program (RDP) RDP4 software. Recombination events between viruses were supported in regions of (**a**,**b**) pregotovirus open reading frame ORF1a, and (**c**) Burmese python septovirus ORF1b.

**Table 1 viruses-16-00310-t001:** Serpentoviral ORF characterization. The serpentovirus genus, host species, and the ORF name are listed along with modeled membrane typology using the PredictProtein server. The closest UnitProtKB protein matches as determined by HMMER for each ORF are listed along with support values, amino acid similarity, and a basic description of matched protein function. Proteins with no close matches were also modeled with ITASSER servers but did not produce close protein matches.

*Genus* or Virus (*Genus*)	ORF	Topology	N-Glyco. Sites	Protein Match	E-Value & % AA Similarity	Matched Protein Name and Function (Origin)
Most Serpentoviruses	TM1_(3)_	2xTM	0.73	-	-	-
*Pregotovirus*	TM2_(6)_	TM+SP	0.77 *	A0A4X2KKC5	0.051 56.9%	Endothelial cell surface expressed chemotaxis and apoptosis regulator (Wombat; *Vombatus ursinus*)
*Pregotovirus, Sectovirus*	GP1_(7)_	TM+SP	0.70	-	-	-
*Lyctovirus*	GP2_(6/7)_	TM+SP	0.77 *	A0A0D0IPT2	0.0094 ** 47.5%	Immunoglobulin (Ig) domain-containing protein (Bacteria; *Leucobacter komagatae*)
MG600029 Chinese Water Snake (*Lyctovirus*)	VP7_4_	TM	-	A0A210PPM4	1.1 × 10^−5^ ** 51.1%	MICAL-like protein 2: Cytoskeleton reorganization & transport (Scallop; *Mizuhopecten yessoensis*)
MG600030 Red Banded Snake (*Lyctovirus*)	VP13_4_	2xTM	0.71	L8WHF5	2.8 × 10^−17^ ** 75.0%	TPR_1 domain-containing protein: Tetratricopeptide repeat domain for Multi-protein scaffolding (Fungus; *Thanatephorus cucumeris*)
MT997160 Veiled Chameleon Serpentovirus A (*Lyctovirus*)	VP54_3_	TM	0.78 *	A7TSS1	0.053 48.3%	Dihydrolipoyllysine-residue succinyltransferase: Acyltransferase enzyme (Fungus; *Vanderwaltozyma polyspora*)
	VP33_6_	2xTM	0.61	A0A1I7VSU5	0.03 ** 58.6%	C2H2-type domain-containing protein: Zinc-finger domain-containing protein (Eye worm; *Loa loa*)
	VP12_7_	-	-	-	-	-
	VP7_8_	-	0.70	A0A6P3ZZ82	0.0014 ** 60.0%	TMV resistance protein N-like: Tobacco-Mosaic Virus resistance/avirulence protein (Plant; *Ziziphus jujuba*)
MT997159 Veiled Chameleon Serpentovirus B (*Vebetovirus*)	VP12_3_	2xTM	-	A0A4Y6I7I9	0.00071 ** 58.1%	Calx-beta domain-containing protein: Calcium binding and expulsion (Plant; *Shewanella sp*.)
	VP34_4_	TM+SP	0.75 *	-	-	-
MG600031 Mandarin Rat Snake (*Sectovirus*)	VP38_4_	TM+SP	0.75 *	H9GBL5	5.5 × 10^−6^ ** 62.5%	Uncharacterized protein: *Anolis* lizard transcription factor (Green anole; *Anolis carolinensis*)
NC043490 Zinzhou Snake Nematode (*Sectovirus*)	VP43_4_	TM+SP	0.77 *	A0A0D3GH98	0.0051 ** 47.7%	Peroxidase enzyme (Plant; *Oryza barthii*)
*Sectovirus*	VP37_8_	TM+SP	0.70	-	-	-
[OR131603-605] Blood Python (*Septovirus*)	VP8_6_	-	-	-	-	-
Other Python *Septoviruses*	VP13_6_	-	-	-	-	-
MG600028 Popes Keelback (*Infratovirus*)	VP7b_4_	TM+SP	-	A0A2A4JQ65	0.0002 ** 60.6%	Erythroid differentiation-related factor 1: Transcription factor for Globin gene (Plant; *Heliothis virescens*)
MN161572 Honduran Milksnake (*Infratovirus*)	VP10_4_	-	-	-	-	-
KX883638 Xinzhou Snake Nematode (*Infratovirus*)	VP13_4_	TM+SP	-	A0A016UTC2	1.7 × 10^−5^ ** 52.2%	Uncharacterized protein (Nematode; *Ancylostoma ceylanicum*)
NC027199 Bovine Nido (*Bostovirus*)	VP9_6_	SP	-	-	-	-

* Highest Glycosylation Potential ≥ 0.75. ** Statistically Significant E-Value ≤ 0.05. TM: one transmembrane helix with extracellular regions. 2xTM: two transmembrane helixes sandwiching extracellular regions. TM+SP: one transmembrane helix with an extracellular region containing a signal peptide. SP: one region of signal peptide.

## Data Availability

Assembled genomic sequence generated from Sanger and Illumina MiSeq sequencing can be found in Genbank accession numbers [OR131594-OR131648] and raw reads from Illumina MiSeq sequencing in Sequence Read Archive accession number [PRJNA982273]. Fragmented genomes were mapped to their closest existing published genomes determined by NCBI BLASTN and putative gaps were mapped in a composite alignment sequence using Geneious Prime. Assemblies for fragmented genomes with putative gaps mapped using the closest published genome can be found at [33].

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
