# Peer review of "Serpentoviruses Exhibit Diverse Organization and ORF Composition with Evidence of Recombination"

_viruses, 2024, doi:10.3390/v16020310_

Round 1

Reviewer 1 Report

Comments and Suggestions for Authors

Very valuable and very interesting publication on the organzation of the serpentovirus genome with new insights.

One minor remark:

2.1 Sample origin and diagnostic screening
Please provide data to species and location of the 11 novel serpentoviruses at this place. They are named lateron (for example, in Figure 2), but it would be helpful to know about species and origin before ( samples from animals from one collection or different collections ? different parts of the USA ?)

Author Response

Very valuable and very interesting publication on the organization of the serpentovirus genome with new insights.

The authors would like to the thank Reviewer 1 for their time, positive comments, and have addressed their comments below.

One minor remark:

2.1 Sample origin and diagnostic screening
Please provide data to species and location of the 11 novel serpentoviruses at this place. They are named lateron (for example, in Figure 2), but it would be helpful to know about species and origin before ( samples from animals from one collection or different collections ? different parts of the USA ?)

To help clarify this, the authors have added to Line 103: Ten novel serpentovirus genomes generated for this study originated from diagnostic samples of eight captive python species from multiple collections within the United States submitted to the Zoological Medicine Diagnostic (ZMDx) Laboratory at the University of Florida.

Additionally, we have created a new Supplemental table (S1), mentioned on Line 143 that includes GenBank accession numbers, host species, reference genome used to map genome gaps, number of sequenced base pairs, and the estimated portion of the total genome represented by sequencing for the eleven novel genomes in the study.

Reviewer 2 Report

Comments and Suggestions for Authors

The authors present the genomic comparison of 11 novel Serpentoviruses and known genera of the subfamily Serpentovirinae. While there comparison is very comprehensive at times it is hard to follow the flow of the manuscript.

Method section:

-      The authors report 11 novel Serpentoviruses but provide 54 accession numbers. They authors should specifiy if coding sequencing for ORFs were submitted independently and supply at least a supplementary table

-      Information should be given how Figure 1 was generated besides the phylogenetic tree on the left hand side

-       

Result section:

-      The authors mention that in some cases only fragmented genomes were generated. The authors should give additional information on the percentage of genome coverage by those partial genomes

-      The authors try to link known proteins with the ORF position in the respective genome by using supercript numbers. However, in some cases those numbers are missing. E.g TM1/2 and GP1/2 in table 1, line 325, line 328, line 342, etc. I would advise the authors to be more consistent.

-      The authors use e-values to refer to high/low-similarity matches in databases (eg lines 308, 310, 327, 366). However, e-values are "expect values" and refer to the probability of a match being random or significant. For those matches it would actually be helpful to also have the similarity to those proteins mentioned as well as or instead of the e-value.

-      The authors only provide information of the likelihood of N-glycosylation sides. Does that only refer to one side or to multiple. Are the sides conserved throughout the genera or highly variable?

-      When describing the 5’ and 3’ UTRs the authors only mention the genomes with the longest UTR sequences available. The shorted one should be included as well (eg. Line 260 and 266)

Author Response

The authors present the genomic comparison of 11 novel Serpentoviruses and known genera of the subfamily Serpentovirinae. While there comparison is very comprehensive at times it is hard to follow the flow of the manuscript.

The authors would like the thank Reviewer 2 for their time and suggestions for improvements on clarity, and have addressed their comments below.

Method section:

The authors report 11 novel Serpentoviruses but provide 54 accession numbers. They authors should specifiy if coding sequencing for ORFs were submitted independently and supply at least a supplementary table

The authors have created a new Supplemental table (S1), mentioned on Line 143 that includes the GenBank accession numbers for sequence fragments, host species, reference genome used to map genome gaps, number of sequenced base pairs, and the estimated portion of the total genome represented by sequencing for the eleven novel genomes in the study to better clarify the novel genomes produced in the study.

Information should be given how Figure 1 was generated besides the phylogenetic tree on the left hand side

To clarify this, the authors have added to the Figure 1 legend,Line 274: Genome templates were created by identifying potential Open Read Frames (ORFs) and grouping common ORFs via Multiple Alignment using Fast Fourier Transform (MAFFT)alignments.

Result section:

The authors mention that in some cases only fragmented genomes were generated. The authors should give additional information on the percentage of genome coverage by those partial genomes

To address this, the following was added to Line 240: MiSeq NGS successfully generated large portions of genomic sequence (estimated 48-97% of genomes represented, Supplemental Table 1) for 11 novel viruses included in this study that fit within existing serpentovirus genera clades

The authors try to link known proteins with the ORF position in the respective genome by using supercript numbers. However, in some cases those numbers are missing. E.g TM1/2 and GP1/2 in table 1, line 325, line 328, line 342, etc. I would advise the authors to be more consistent.

The authors have added subscript numbers throughout the text to more consistently reference protein ORF numbers.

The authors use e-values to refer to high/low-similarity matches in databases (eg lines 308, 310, 327, 366). However, e-values are "expect values" and refer to the probability of a match being random or significant. For those matches it would actually be helpful to also have the similarity to those proteins mentioned as well as or instead of the e-value.

The authors have added amino acid similarity to Table 1 as well as throughout the text where E-values are mentioned.

The authors only provide information of the likelihood of N-glycosylation sites. Does that only refer to one site or to multiple. Are the sites conserved throughout the genera or highly variable?

The N-glycosylation sites reported in this study are only the site with the highest predicted likelihood for each protein. While the presence of N-glycosylation sites was included in our analysis, the specific position of those sites across genomes was not covered in our analysis. To address these limitations, the authors have added the following:

Line 196: For proteins with multiple predicted N-linked glycosylation sites, only the site with the highest likelihood was reported in the analysis.

Line 621: Although many proteins had multiple predicted N-linked glycosylation sites, only the site with the highest likelihood was reported in the analysis

When describing the 5’ and 3’ UTRs the authors only mention the genomes with the longest UTR sequences available. The shortest one should be included as well (eg. Line 260 and 266)

The authors have included the minimum and maximum lengths recorded for UTR’s by including:

Line 266: The length of the 5’ UTR varied from 289 bp up to 2271 bp

Line 273: [3’UTR] Lengths for this region varied from 165 bp to 1653 bp in length